# Technological Approaches for Improving Vaccination Compliance and Coverage

**DOI:** 10.3390/vaccines8020304

**Published:** 2020-06-16

**Authors:** Céline Lemoine, Aneesh Thakur, Danina Krajišnik, Romain Guyon, Stephanie Longet, Agnieszka Razim, Sabina Górska, Ivana Pantelić, Tanja Ilić, Ines Nikolić, Ed C. Lavelle, Andrzej Gamian, Snežana Savić, Anita Milicic

**Affiliations:** 1Institute of Pharmaceutical Sciences of Western Switzerland, University of Geneva, Rue Michel-Servet 1, 1221 Geneva, Switzerland; celine.lemoine@unige.ch; 2Vaccine Formulation Institute, Chemin des Aulx 14, 1228 Plan-les-Ouates, Switzerland; 3Department of Pharmacy, Faculty of Health and Medical Sciences, University of Copenhagen, Universitetsparken 2, 2100 Copenhagen Ø, Denmark; aneesh.thakur@sund.ku.dk; 4Department of Pharmaceutical Technology and Cosmetology, University of Belgrade-Faculty of Pharmacy, Vojvode Stepe 450, 11221 Belgrade, Serbia; danina.krajisnik@pharmacy.bg.ac.rs (D.K.); ivana.pantelic@pharmacy.bg.ac.rs (I.P.); tanja.ilic@pharmacy.bg.ac.rs (T.I.); ines.nikolic@pharmacy.bg.ac.rs (I.N.); snezana.savic@pharmacy.bg.ac.rs (S.S.); 5The Jenner Institute, Nuffield Department of Medicine, University of Oxford, Old Road Campus Research Building, Roosevelt Drive, Oxford OX3 7DQ, UK; romain.guyon@ndm.ox.ac.uk; 6Virology & Pathogenesis Group, Public Health England, Manor Farm Road, Porton Down, Salisbury SP4 0JG, UK; stephanie.longet@phe.gov.uk; 7Department of Microbiology, Hirszfeld Institute of Immunology and Experimental Therapy, Polish Academy of Sciences, ul. Rudolfa Weigla 12, 53-114 Wroclaw, Poland; agnieszka.razim@hirszfeld.pl (A.R.); sabina.gorska@hirszfeld.pl (S.G.); 8The Adjuvant Research Group, School of Biochemistry and Immunology, Trinity Biomedical Sciences Institute, Trinity College Dublin, DO2R590 Dublin, Ireland; lavellee@tcd.ie; 9Department of Immunology of Infectious Diseases, Hirszfeld Institute of Immunology and Experimental Therapy, Polish Academy of Sciences, ul. Rudolfa Weigla 12, 53-114 Wroclaw, Poland; andrzej.gamian@hirszfeld.pl

**Keywords:** vaccine delivery, compliance, microfluidics, mucosal vaccination, cutaneous vaccination, adjuvants

## Abstract

Vaccination has been well recognised as a critically important tool in preventing infectious disease, yet incomplete immunisation coverage remains a major obstacle to achieving disease control and eradication. As medical products for global access, vaccines need to be safe, effective and inexpensive. In line with these goals, continuous improvements of vaccine delivery strategies are necessary to achieve the full potential of immunisation. Novel technologies related to vaccine delivery and route of administration, use of advanced adjuvants and controlled antigen release (single-dose immunisation) approaches are expected to contribute to improved coverage and patient compliance. This review discusses the application of micro- and nano-technologies in the alternative routes of vaccine administration (mucosal and cutaneous vaccination), oral vaccine delivery as well as vaccine encapsulation with the aim of controlled antigen release for single-dose vaccination.

## 1. Introduction

Estimated global immunisation coverage varies per country and per vaccine, with some vaccines such as pneumococcal and rotavirus ranging between 35–47%, falling significantly short of the WHO’s recommended coverage of >90%. Although many different factors affect these figures, vaccination compliance is one of the major obstacles to broader immunisation coverage globally. Pain that is felt during parenteral vaccine administration is one of the most frequently reported causes of vaccine hesitancy. Success of the oral polio vaccine in bringing the disease close to eradication (notwithstanding recent issues related to the vaccine itself [1]), and the increasing uptake of the nasal flu vaccine for children, indicate that non-injectable routes of vaccination have a significant impact on improving coverage and compliance. Vaccines also have a critical role in slowing down and preventing the spread of antibiotic resistance worldwide [2]. Approaches to broaden vaccination coverage include the development of new technologies for the mode and route of vaccine administration, and formulations for cutaneous, oral and mucosal delivery (Figure 1). We discuss here the targeted strategies and key innovations of different vaccination technologies, highlighting their potential to overcome common vaccination-related limitations and improve compliance.

## 2. Vaccine Formulations for Pulmonary and Nasal Delivery

Mucosal immunity is the first and foremost line of defence against pathogens that enter the body through the oral mucosa and the nasal cavity, such as influenza and viruses causing meningitis, measles or whooping cough. Estimates of the surface area of the mucus membranes lining the lungs vary between 50–75 square meters, representing the largest epithelial surface exposed to the outside environment. However, most licensed vaccines are administered by intramuscular injection, which preferentially induces systemic immune responses [3]. Protective mucosal responses can be effectively elicited by mucosal immunisation [4]; furthermore, mucosal vaccines are attractive by being non-invasive and needle-free.

The airway mucosa is the site of substantial immunological activity where constant immune monitoring recruits highly professional innate and adaptive immune cells, to protect the host from microbial and environmental insults. Nasopharynx-associated lymphoid tissue (NALT) is an organised mucosal immune system that consists of lymphoid tissue, B cells, T cells and antigen presenting cells (APCs), covered by an epithelial layer containing microfold (M) cells [5]. M cells in the epithelial cell layers have a specialised role in transporting antigens across the epithelium [5,6]. Following mucosal administration, immunity is induced at mucosal as well as serosal surfaces and in both local and distal mucosa. Compared to conventional injectable vaccines, mucosal vaccines have many other advantages: ease of administration, better patient compliance, lower costs, avoidance of needle stick injuries and needle waste, and the scope for mass immunisation [7]. The interconnection of various mucosal sites through a common mucosal immune system allows the possibility to immunise via the nose against infectious diseases that originate from distal mucosal sites [8].

Intranasal administration of a vaccine allows the induction of a strong systemic and local immune response. In a recent phase I clinical trial of a novel intranasal respiratory syncytial virus (RSV) F protein vaccine, which was linked to an immunostimulatory bacterium-like particle, persistent nasal IgA and serum IgG responses were observed for up to six months [9]. In addition, the enzymatic activity in the nasal cavity is relatively weak, compared to the oral route, which can be a reason why some vaccines are more efficient when administered intranasally. In a study that compared the mucosal delivery of the intranasal and oral *Bordetella bronchiseptica* vaccine, the intranasal vaccine conferred stronger clinical immunity [10]. However, antigens can also be rapidly removed from the nasal cavity or poorly absorbed by epithelial cells, potentially leading to reduced immunogenicity [11]. Nasal vaccines can be delivered in the form of powder, aerosol, gel or drops. Each of these forms have their advantages and disadvantages, as reviewed previously [12].

Mucosal vaccination can protect against several mucosally transmitted bacterial and viral diseases, and a few oral and nasal mucosal vaccines have been authorised for use in humans e.g., sabin polio, rotavirus, and nasal influenza vaccine [7]. All of these mucosal vaccines are based on the entire pathogen, either killed or live attenuated. Thus, they are associated with many disadvantages including laborious and expensive production and distribution, and the risk of reversion to the virulent form.

Conversely, the new generation of subunit vaccines that are based on single or multiple highly purified pathogenic antigens, such as peptides, proteins, polysaccharides, and nucleic acids, represent safer alternatives. Such subunit vaccines are inherently poorly immunogenic and require the inclusion of an adjuvant. This is especially relevant for mucosal delivery routes where the targeted mucosal epithelium, which naturally is in contact with many possible antigens, requires a strong immune-potentiating signal in order not to induce tolerance [13]. Subunit vaccines for pulmonary and nasal immunisation, when combined with adjuvants, can overcome many of the shortcomings of conventional vaccines. Table 1 summarises the vaccines delivered through pulmonary and nasal routes in different phases of clinical testing.

Adjuvants comprise structurally diverse compounds, which can be categorised as delivery systems or immunopotentiators, or a combination of both [14]. Delivery systems enhance the immune response against the co-delivered antigen by protecting it from degradation and allowing sustained antigen release. Different classes of delivery systems that can be used as a platform technology for delivering vaccines include liposomes, polymer particles, inorganic particles, bacterial or viral vectors, outer membrane vesicles, immunostimulating complexes, emulsions, and virus-like particles (Table 2). These vaccine carriers can extend the residence time of the antigen on the mucosa, which increases its chances of getting into the deeper layers and reaching the immune cells [15]. Different strategies can be used to deliver vaccine antigen to the mucosa: mucoadhesion, M cell targeting or targeting APCs. Mucoadhesion can be achieved by employing positively charged carriers such as chitosan or liposomes.

Endocine™ is a mucosal adjuvant based on endogenous lipids found in the human body that possess a negative charge. It was shown to be both safe and effective in inducing a humoral and cell-mediated immune (CMI) response after intranasal administration in animal models, including in aged mice [16]. A positively charged oil-in-water nanoemulsion, when combined with the H5 hemagglutinin antigen, was recently shown to be effective in protecting against influenza challenge in ferrets [17]. A carrier composed of liposomes covered with chitosan induced both systemic and mucosal immunity in mice when administered intranasally together with an epitope from group A *Streptococcus* protein [18]. There are many specific targets on the M cell surface that can be used for designing an efficient vaccine carrier [19]. Khan et al. used a conjugate of GB-1 M cell ligand and F and G protein fragments from RSV to vaccinate mice intranasally and reported an efficient mucosal and systemic response, as well as protection against nasal challenge with RSV [20]. The conjugation of chitosan with mannose increases its engulfment by dendritic cells (DCs) and macrophages, as these cells express the mannose receptor on their surface. Mannosylated chitosan used in a DNA vaccine against *M. tuberculosis* administered intranasally to mice-induced secretory IgA (sIgA) in the broncho-alveolar lavage (BAL) fluid and provided improved protection in challenge experiments [21]. Another example is the use of β-glucan which is recognised by the Dectin-1 receptor localised on DCs [22]. The addition of β-glucan to chitosan-HbsAg vaccine significantly increased the anti-HbsAg antibody titre in immunised mice [23]. It has to be underlined that in some cases the exact mechanism of APC targeting is unknown. DOTAP/DC-chol liposomes administered intranasally with pneumococcal surface protein A provided protection against pneumococcal infection, and were shown to be specifically engulfed by DCs, even though these particles do not possess any known DC ligands [24].

Immunopotentiators activate the immune system through pattern-recognition receptors (PRRs) expressed by APCs. Immunopotentiators can be bacterial or viral toll-like receptor (TLR) agonists, stimulator of interferon genes (STING) agonists, and cytokines (Table 3). Delivery systems and immunopotentiators together determine the magnitude and quality of the innate immune response and the subsequent adaptive immune response specific to the co-delivered vaccine antigen. Vaccine antigens are delivered to dendritic cells, the most specialised APCs, initiating the differentiation of T-helper cell subsets, which in turn interact with B cells, eventually resulting in the production of antibodies (sIgA) at mucosal sites [8].

### Dry Powders for Pulmonary Immunisation

Most vaccine antigens are macromolecules, such as polysaccharides, proteins, peptides, and nucleic acids, and usually are at a great risk of chemical and physical degradation in liquid formulations [25]. All vaccines lose potency over time, and the rate of potency loss is dependent on the handling and storage temperature. The delivery of vaccine antigens in the form of dry powder particles to the lungs is recognised as a potential immunisation strategy that improves vaccine stability in comparison to liquid vaccine formulations [26]. The thermostability of vaccine antigens can be further improved by formulating them as dry powder microparticles in the presence of sugars as stabilising excipients [26]. A number of drying methods such as spray-drying, freeze-drying, and spray freeze-drying are used to prepare dry powder vaccine particles [26]. These approaches to develop thermostable vaccine formulations that are resistant to damage caused by freezing or overheating also eliminate the dependence on a cold chain. Thus, dry powder-based inhalable vaccine formulations for pulmonary immunisation not only induce systemic and mucosal immune responses [8], but also have logistical advantages over injectable vaccines [26]. Dry powder-based vaccine formulations have been designed and pre-clinically tested against several infectious diseases, progressing into clinical trials (Table 1). An inhalable vaccine formulation of alginate-coated live *Mycobacterium* microparticles was more immunogenic than liquid aerosols, and provided better protection in mice against experimental *Mycobacterium tuberculosis* infection [27]. In another study, an intrapulmonary-delivered Advax-adjuvanted influenza vaccine induced higher memory B and T cell responses than intranasal or intramuscular immunisation and conferred superior disease protection [28]. The development of inhalable dry powder vaccines is thus a promising new strategy for pulmonary immunisation. However, a number of parameters defines the optimal performance of dry powder vaccines such as aerodynamic particle size, aerosolisation performance, antigen stability, controlled release, drug delivery device, safety, and the scale-up of manufacturing. Advancements in pharmaceutical and nano-technologies enabling the development and testing of dry powder vaccines for pulmonary immunisation should help to lay the groundwork for the successful commercialisation of the first aerosolised mucosal vaccine.

## 3. Oral (Gastrointestinal) Vaccines

Oral delivery is the most patient-friendly route of administration, and consequently, oral vaccines have the potential to improve vaccine efficacy by enhancing their accessibility and distribution, which may lead to better vaccine coverage [3]. Oral vaccination is also regarded as the optimal means to fight infections caused by enteric pathogens as it induces intestinal immunity through the gut-associated mucosal tissues [62]. The first successfully implemented oral vaccine was the oral polio vaccine developed in the 1950s by Albert Sabin. It had the ability to induce protective sIgA responses in the intestinal mucosa, the main site of poliovirus entry and multiplication. This significantly reduced viral transmission, leading towards the global eradication of polio [8]. Other licensed oral vaccines target diseases induced by enteric pathogens such as *Vibrio cholerae*, *Salmonella typhi* and rotavirus, causing cholera, typhoid fever and gastroenteritis, respectively [63].

Despite the clear benefits of oral vaccines, only a few have been successfully developed. Oral vaccines have to overcome difficult challenges linked to the gastrointestinal biology: the acidic environment in the stomach, the proteolytic enzymes necessary for protein degradation, the presence of mucus, low intestinal permeability and the generally poor immunogenicity of orally delivered antigens [64]. Consequently, an efficient oral vaccine should ideally be (1) stable in a highly enzymatic environment and resistant to site-specific pH; (2) delivered to specific immune-induction sites (e.g., Peyer’s patches in the small intestine); (3) adapted to interactions with mucus; (4) able to be transported through the intestinal epithelial barrier; (5) captured by appropriate APCs and able to evade mucosal tolerance [19]. A number of oral drug delivery technologies are currently under pre-clinical and clinical development to overcome these challenges [65]. Various particle-, liposomal- or adenoviral-based systems have been evaluated as vehicles to deliver the vaccine antigens [62]. Other promising formulations are site-specific delivery systems, which are often capsule- or tablet-based. These systems can facilitate antigen protection and the delivery of vaccines to specific areas of the gastrointestinal tract and particularly to the key sampling sites such as the Peyer’s patches [19]. Site-specific release can be achieved through the application of pH-dependent coating such as shellac, cellulose acetate phthalate, cellulose acetate trimellitate, poly(vinyl acetate phthalate), or hydroxypropyl methylcellulose phthalate [66]. These delivery systems play an important role in the stability and the delivery efficacy of vaccine components to the intestine. However, additional targeted strategies to specifically induce intestinal immune responses may be beneficial.

Ligand-mediated vaccine delivery systems have been shown to direct antigens to specific receptors expressed on intestinal M cells, epithelial cells or intestinal APCs [19]. Given the pivotal role of M cells in antigen sampling, several lectin-, antibody- and peptide-based targeted strategies have been developed to specifically engage these cells. Plant lectin-based ligands, facilitating the bioadhesion to glycans expressed on M cells, have been tested. *Ulex europaeus* agglutinin-1 (UEA-1) was shown to specifically bind to α-L-fucose residues expressed on mouse Peyer’s patch M cells [67,68] and able to target polystyrene microparticles [69] or liposomes [70] to them. Since many intestinal pathogens gain entry to the host through M cells, some M cell receptors used by bacteria have also been evaluated. For example, glycoprotein 2 (GP2) is expressed on murine and human intestinal M cells, and *Escherichia coli* (*E. coli*) and *Salmonella enterica typhimurium* are able to bind GP2 through FimH, a pili component on the bacterial outer membrane [71]. The conjugation of an anti-GP2 monoclonal antibody to ovalbumin (OVA) resulted in effective M cell targeting and oral immunisation with this system triggering enhanced faecal OVA-specific sIgA responses compared to the antigen alone in mice [72]. Other M cell-specific antibodies have been analysed. The conjugation of the anti-M cell antibody 5B11 to polystyrene particles enhanced their uptake by rabbit intestinal M cells in an ileal loop model [73], while oral vaccination with a conjugate of the NKM 16-2-4 antibody to botulinum toxoid (BT) enhanced BT-specific serum IgG and mucosal IgA responses as well as protective immunity against lethal challenge with BT in mice [74]. Finally, some peptides targeting M cells have been tested by oral route. The tripeptide Arginine-Glycine-Aspartic Acid (RGD) motif, which could bind to β-integrins on M cells, was shown to enhance antigen-specific serum IgG responses in mice [75] and the tetragalloyl-D-lysine dendrimer (TGDK) targeting murine, human, and nonhuman primate M cells was demonstrated to enhance faecal antigen-specific IgA responses in macaques [76].

In addition to targeted strategies to engage specific cells, including M cells, the choice of antigen and adjuvant is pivotal in developing efficient oral vaccines. The current licensed oral vaccines are composed of either live-attenuated or killed organisms, sometimes in combination with protein subunit components [77]. To enhance the antigen immunogenicity, novel recombinant enterotoxigenic *E. coli* (ETEC) [78] and *Vibrio cholerae* [79] strains, overexpressing antigens or expressing multiple antigens, have been developed and were successfully tested in animal models and in clinical trials. Considering that much vaccine development is currently focused on subunit vaccines, the addition of adjuvants to the formulation may be crucial to overcome intestinal tolerance. However, there are currently no licensed adjuvanted oral vaccines for human use.

Cholera toxin (CT) and the heat-labile enterotoxin of *E. coli* (LT) have been shown to be potent mucosal adjuvants in pre-clinical studies. However, the native forms of these toxins are too toxic for use in humans, leading some research groups to develop toxin or toxin subunit mutants [77]. For instance, double-mutant labile toxin (dmLT) has been tested in pre-clinical studies [80] and evaluated in a Phase I clinical trial as part of a prototype oral ETEC vaccine [81]. Interestingly, dmLT was shown to promote Th17 responses but also protective sIgA responses in pre-clinical studies [82]. A non-toxic CT derivative was also developed as CTA1-DD which is a fusion between the A subunit of CT and the D-fragment of the *Staphylococcus aureus* protein A [83]. CTA1-DD was shown to be safe and enhanced the immunogenicity of various antigens in animal models [84].

Unconventional T cells such as invariant natural killer T (iNKT) cells or mucosal-associated invariant T (MAIT) cells have been considered as potential adjuvant targets: Intestinal sites are enriched in these cells which are at the interface between innate and adaptive immunity and can modulate APCs [85]. Some iNKT cell agonists have been investigated as adjuvants to enhance immune responses in immunotherapy and vaccination strategies [86]. Lavelle and colleagues demonstrated the potential of the iNKT cell activator α-Galactosylceramide (α-GalCer) as an oral adjuvant to enhance intestinal immune responses induced by experimental whole-cell killed ETEC [87], *Vibrio cholerae* [88] and *Helicobacter pylori* [89] antigens in mouse models. In addition, a novel oral delivery-integrated system named Single Multiple Pill^®^ (SmPill^®^), containing oil-in-water emulsions formulated as 1 mm minispheres, was reported to effectively protect and enhance the release of various drugs in targeted intestinal regions [90]. The SmPill^®^ integrated system, incorporating a recombinant formalin-killed whole-cell *E. coli* overexpressing the colonisation factor antigen I (CFA/I) and the orally active adjuvant α-GalCer, was shown to facilitate a controlled and sustained antigen release at intestinal pH [87]. Furthermore, this vaccine delivery system was able to enhance the intestinal CFA/I-specific sIgA responses in mice. It was also shown that a novel whole-cell killed *Vibrio cholerae* strain and recombinant cholera toxin subunit B (CTB) could be successfully loaded as antigens in SmPill^®^ minispheres. Consistent with the previous findings, Davitt and colleagues demonstrated that combining these antigens and α-GalCer in SmPill^®^ minispheres enhanced intestinal lipopolysaccharide and CTB-specific IgA responses and induced intestinal antigen-specific Th1 responses (Figure 2) [88].

Finally, a key concern in vaccine formulation is stability. Indeed, the development of thermostable vaccine formulations may improve vaccine coverage, especially in low-income countries. The stability of this integrated oral delivery system containing formalin-killed whole-cell *E. coli* overexpressing CFA/I, and the orally active adjuvant α-GalCer, was evaluated under various temperature and humidity conditions. Longet and colleagues determined that SmPill^®^ minispheres maintained both the antigenicity of CFA/I and the immunostimulatory activity of the α-GalCer adjuvant after the storage of SmPill^®^ minispheres under room temperature and extreme storage conditions for several months [91]. Collectively, these results support the potential of the SmPill^®^ minisphere approach to enhance the immunogenicity of orally delivered antigens [87] and maintain the stability of oral vaccine formulations [91]. This exemplifies the potential to use integrated strategies to overcome challenges in developing oral vaccines.

## 4. Cutaneous Immunisation

Skin or (trans)cutaneous vaccination is a route of immunisation mediated by topical, intradermal (ID) or intraepidermal delivery [92]. In line with global aspirations to expand vaccination coverage, cutaneous vaccines are regarded as a promising option for overcoming diverse issues, from vaccine safety and reactogenicity to patient preferences [93]. As an immunocompetent and multi-functional organ, the skin appears to be highly susceptible to certain vaccine adjuvants, resulting in enhanced immunogenicity and allowing the reduction in antigen dose and immunisation frequency [94]. Therefore, it is important that the immunogenicity profiles of cutaneous vaccines are not significantly inferior to other routes of vaccination [93]. Among the many technological approaches for cutaneous vaccination, several deserve a more detailed overview.

Electroporation is an electro-permeabilisation method, being intensively explored in different fields: DNA manipulation in vitro, drug delivery and gene therapy. It is based on the transitory structural perturbation of lipid membranes (such as the cell membrane) through the application of high-voltage electrical impulses. Typically, electroporation involves the short-term exposure (few µs to ms) to high-voltage pulses (50–1500 V) with up to 1 s intervals. It is hypothesised that a structural rearrangement in the lipid bilayer occurs, consequently forming transient pores and facilitating molecular transport for both small molecules and biologics [95]. In the context of skin vaccine delivery, the CELLECTRA^®^ electroporation device (developed by Inovio Pharmaceuticals) has shown good safety and efficacy [96] and is currently being assessed in a Phase III clinical study for DNA-based immunotherapy of cervical cancer (NCT03721978). It has more recently also been applied to a DNA-based coronavirus vaccine (NCT04336410). Interestingly, there are findings suggesting that electroporation alone may act as a physical adjuvant, by stimulating the ‘trickling’ of Langerhans cells (LCs) away from the treatment site (presumably to lymph nodes) and inducing a certain level of pro-inflammatory cytokines [97].

Among the different thermal microporation technologies, fractional infrared laser ablation stands out as particularly suitable for bypassing the skin barrier properties. It enables the disruption of the stratum corneum in a highly controlled and adjustable fashion, simultaneously providing an intrinsic adjuvanting effect. Consequently, this technology has been intensively investigated for the pain-free prophylactic and therapeutic treatment of type I allergies and tumours [92,98,99]. Despite a two-step administration process (microporation followed by antigen application), laser ablation could potentially be valuable in mass vaccination campaigns, particularly in combination with dry vaccine patches, offering advantages such as heat stability, the avoidance of needle-related injuries, and improved uptake, at a cost similar to conventional vaccination [99]. Currently, the major challenge is how to standardise the pore depth and ensure sufficient reproducibility, due to the variability of skin thickness depending on body site, age and ethnicity [92]. The available results encourage further investigations in healthy adults to evaluate the safety and efficacy of laser microporation prior to, for example, vaccine patch application [98].

Microneedle arrays represent another interesting approach for intradermal vaccine delivery. Although these devices comprise needles, their length (10–2000 μm) offers relatively pain-free application [100]. Among the many microneedle types, future widespread vaccine delivery is expected for solid and dissolvable needles [93,101]. However, attaining reproducible coating and mechanical properties remain among the frequently noted critical attributes. Preclinical studies conducted to date imply that microneedles may generate immunogenicity comparable to intradermal or intramuscular vaccination, with some studies reporting higher and more durable antibody and cellular responses [101]. Satisfactory immune responses may be attained even without adjuvant addition or with considerably lower adjuvant doses than otherwise required, which is an important safety asset [92]. Therefore, microneedle patches are relatively cost effective, easy to produce and accepted well by the patients [102]. The fact that they allow self-administration may be the most significant factor in the further prospects for this mode of vaccine delivery [101].

In the search for an improved needle-free vaccination strategy, the application of dry vaccine powders using ballistic (powder) injectors seems to be a promising approach for delivering antigens to the skin, owing to improved vaccine stability and cold-chain independent logistics [103,104]. As a result, this technology has been recognised as potentially suitable for mass vaccination campaigns in developing countries [104]. Numerous studies show that DNA and RNA vaccines, as well as conventional vaccines in a dry state could be administered using the powder injectors [92]. However, in order to achieve the successful delivery of a particulate vaccine into the skin, apart from the design of ballistic devices, the powder properties (composition, particle size, shape, density) have to be carefully adjusted [104]. It should be emphasised that, although adequate immune responses have been observed in numerous preclinical and clinical studies, there are currently no ballistic injection products authorised for human use. The main unsolved issues are the high cost, cutaneous adverse reactions (e.g., erythema, petechiae, skin discoloration, oedema and skin flaking) and pain, that could lead to reduced patient adherence [105,106].

Jet injection is another needle-free approach that delivers liquid vaccine formulations in a 2–500 μL range using a highly pressured propulsion system connected to a needle-free syringe or nozzle [92,107]. The delivery of the vaccine can occur intradermally but also subcutaneously or even intramuscularly, forming a depot, depending on the applied velocity and overall design. Apart from a variation in the depth of delivery, other safety-related issues are successfully circumvented by the fact that this method now utilises prefilled disposable delivery units, thus avoiding contamination. If future research focuses on achieving more cost-effective manufacturing, jet injectors may be a part of promising multi-platform systems (e.g., coupled with nanoparticulate formulations) for driving further the cutaneous vaccination approach [107,108].

The aforementioned physical devices, although efficient in antigen delivery into the skin, may lead to some skin barrier damage, making them less suitable for mass vaccination under critical hygienic conditions [108,109]. As a result, there has been an increasing interest in passive delivery strategies, particularly nanocarriers, enabling antigen application to intact skin as well as improved antigen stability, sustained antigen release and increased antigenicity by mimicking the size of microorganisms [108,110,111]. Until now, different nanoparticles have been studied for this purpose, including vesicular nanocarriers (transfersomes, ethosomes, liposomes, niosomes, nanoemulsions) and solid nanoparticles (polymeric nanoparticles, silica-based nanoparticles) [92,110]. However, although nanoparticles may lead to a superior immune response compared to conventional intramuscular immunisation (particularly with an appropriate adjuvant), progress towards clinical settings has been negligible, due to difficulties in ensuring the accurate, reproducible and efficient delivery of antigen-loaded nanoparticles into the epidermal and dermal tissue [109,111]. Interestingly, in recent years, vaccination via the follicular route using nanoparticles, particularly non-flexible ones, has been recognised as a promising approach. It offers a CD8^+^ T cell-biased immune response (due to a large number of perifollicular APCs) that could be beneficial for the development of vaccines against intracellular pathogens, viruses and cancers [112]. However, it is important to emphasise that research in this area is still at an experimental stage, due to numerous factors affecting trans-follicular immunisation with nanoparticles, including particle size, surface properties (charge and composition of surface layer) and hair cycling [109].

## 5. Controlled Antigen Release Delivery Systems for Single-Dose Immunisation

Since the early 1990s, there has been extensive research on controlled release delivery systems for vaccine applications [113,114]. Prior to this, controlled delivery technologies were developed for a sustained delivery of drug molecules and provided the basic understanding of controlled release applications. This paved the way for delayed drug delivery technologies based on “smart” polymers, such as biodegradable microparticles, solid implants or in-site gel-forming implants [115] and the notion of developing biodegradable microparticles to accommodate prime-boost vaccine regimens within a single-dose formulation [116]. Encapsulating the booster vaccine into polymer particles should enable the pulsatile or continued release of the booster dose, that when combined with a free priming vaccine, can mimic a prime-boost regimen within a single immunisation. This approach would alleviate logistical challenges, the costs and the pressure on resources to deliver the booster doses, leading to increased vaccination compliance and coverage globally. Vaccine encapsulation also addresses the need for antigen sparing by improving immune responses through antigen shielding, controlled antigen release and adjuvanting effect due to the particulate nature of the encapsulated vaccine delivery system.

The biodegradable polymer of choice is the FDA-approved poly lactic-co-glycolic acid (PLGA), which can be formulated into nano- and microparticles using a range of methods, the most common being water-in-oil-in-water (W/O/W) double emulsion solvent evaporation. An advantage of using polymer particles for vaccine delivery is that they can be adapted in size and/or structure to resemble a pathogen or to achieve the targeted delivery to promote humoral and CMI responses [117,118]. Particle size influences the immune response. For example, an efficient phagocytosis by macrophages may induce a more rapid immune response, while larger particles avoid the direct uptake by APCs and prolong antigen release [119,120,121,122]. Encapsulated antigens are protected from enzymatic degradation or rapid elimination in vivo, further contributing to an enhanced immune response [117,118]. The tuneable characteristics of the particulate vaccine delivery systems and the possibility for co-encapsulation or surface attachment of immunostimulatory agents play an important role in achieving the desired immune responses or adjuvant effects [123,124]. Various administration routes showing pre-clinical success have been reported for the oral, intranasal, parenteral and dermal delivery of antigens encapsulated in PLGA particles. These vaccine candidates, formulated using the W/O/W method, show great potential as a versatile vaccine delivery system and can achieve various immunologic requirements such as boosting antibody titres and inducing a CMI response. Although PLGA-based microparticles may no longer be a novel approach for vaccine delivery, there is a need for technological innovation to achieve more efficient, cost-effective, preferably solvent-free manufacturing methods of polymer-based controlled antigen-release delivery systems.

### Progress of PLGA Polymer Vaccine Delivery Systems

The main method to achieve controlled antigen release has been the encapsulation into a polymer matrix by the W/O/W method. To this day, it remains the classic method that has been used to produce a wide array of particles with varying characteristics [125]. In general, for W/O/W, the primary emulsion consists of the antigen in an aqueous buffer emulsified into an ‘oil’ phase containing the polymer of choice in a selected solvent. This is followed by a secondary emulsification of the primary emulsion into an aqueous solution containing stabilising surfactants. Lastly, during a solidification phase the particles precipitate and harden as the solvent is evaporated. The manufacturing parameter of this method can be modified at various steps to obtain particles with the desired physicochemical characteristics. PLGA has been the most popular choice of biodegradable polymer and extensive research has demonstrated the influence of the co-polymer ratio, polymer viscosity, molecular weight, volume ratios and end caps on antigen release [116,126,127]. It is important to consider that it is not necessarily the PLGA polymer itself, but the particulate nature of the delivery system that allows for its immune-enhancing activity. The particulates are recognised as foreign material, triggering an immune response involving phagocytosis, the production of cytokines and the further activation of T cells.

The potential of encapsulated vaccines was initially demonstrated using ovalbumin (OVA). Superior IgG responses were detected in mice receiving subcutaneous immunisation of OVA encapsulated in PLGA particles (10 µm) compared to OVA combined with Freund’s adjuvant [113]. Liu et al. demonstrated that more efficient cross-presentation was achieved with OVA adsorbed or encapsulated into lipid-PLGA nanoparticles, exemplifying the potential to use integrated delivery strategies [128]. Numerous studies have confirmed the influence of particle size on both release kinetics and the resulting immune response [119,121]. Studies with smaller particles (0.3–7 µm) [129] and larger particles (100–150 µm) [130] both reported superior IgG titres after intraperitoneal administration. This highlights the difficulty of obtaining direct size and effect correlations for encapsulated vaccine delivery systems. Depending on the antigen and the desired immune response, the optimal particle size may differ greatly. Additionally, differences in the route of administration, animal models, and encapsulation efficiencies should be considered.

Alternative polymer materials have also been explored. Microparticle formulation strategies based on poly (D,L-lactic acid) (PLA) have been investigated in encapsulating various antigens including rotavirus, tetanus toxoid and *Vibrio cholerae* [131]. PLA particles (2–8 µm) with encapsulated tetanus toxoid, administered intramuscularly to Wistar rats, induced high antibody titres. A further increase was observed when co-administered with Alhydrogel adjuvant [120,132]. The incorporation of poly (ethylene glycol) (PEG) was investigated with *Vibrio cholerae*-loaded PLA/PEG microparticles (4–5 µm), inducing high antibody titres as well as protection against a lethal challenge after oral administration in CD-1 outbred mice [133].

The W/O/W method development includes combining PLGA with polysaccharides, or using polysaccharides alone, to formulate antigen delivery systems. A combination of PLGA and sodium alginate investigated for the intradermal delivery of encapsulated malaria peptides demonstrated that particles of 1 µm induced a balanced Th1/Th2 response in BALB/c mice. In combination with immunostimulatory peptides (Arg–Gly–Asp), a strong CMI response was measured compared to the antigen alone or encapsulated in PLGA only [134]. Cationic polymers, such as chitosan, have been explored to formulate particles, however, these require emulsification with internal or external gelation techniques. When combined into PLGA particles, chitosan takes on an adjuvant role due to its inherent positive charge [135]. Novel functionalised dextrans have also been investigated in vaccine delivery systems [136].

The co-delivery of the encapsulated antigen and an adjuvant to enhance antigen presentation has been one of the focal points in the last decade, as characterisation methods have become increasingly advanced. Microparticle formulations containing aluminium salts and TLR9 agonist CpG oligonucleotides have shown that combining the antigen and adjuvant into a particulate delivery system can result in robust immune responses for single-dose vaccines compared to multi-immunisation schedules [124,132,137]. As mentioned above, other approaches have formulated the particles with ligands such as mannose to achieve DC targeting [21,135].

Multi-platform systems such as the single-dose pandemic influenza vaccine of recombinant outer membrane vesicles (rOMVs) encapsulated in PLGA demonstrated equivalent IgG titres to a prime and boost regimen, and protected mice against a challenge with a lethal dose of H1N1, six months post the initial vaccination. These microparticles (10–20 µm), manufactured using W/O/W, had a prolonged antigen release over 30 days [138]. This work demonstrates that significant results can be achieved when combining the classic W/O/W method with novel vaccine antigen technologies.

## 6. Advanced Vaccine Encapsulation Methods

The classic W/O/W emulsification method has a number of shortcomings that have impeded the successful commercial development of an encapsulated antigen in polymer microparticles for single-dose vaccines. Formulation issues and the lack of particle uniformity lead to uneven antigen release profiles, including an initial burst (release of antigen upon injection) [139]. Considerable progress has been made to develop encapsulation technologies that address the issues of antigen stability, encapsulation efficiency, particle size and distribution and the suitable release profiles [140,141].

To encapsulate antigens, stability is critical and the encapsulation method needs to be optimised for each vaccine. Antigen stability may be impaired due to mechanical and chemical stress induced during emulsification steps. Novel encapsulation technologies that use milder processing methods are likely to mitigate the mechanical stress and reduce solvent-surface interactions; one such approach is coaxial electrospraying [142]. Furthermore, antigen stability can be achieved with the co-encapsulation of stabilising additives such as hydrophilic PEG, surfactants, sugars, or protein serum albumin [131,133,142]. The co-encapsulation of cationic excipients such as Eudragit E, poly(L-lysine) and branched polyethylenimine has shown promising results for inactivated polio vaccine encapsulation by W/O/W into PLGA microparticles [143].

The very adaptable process parameters of the W/O/W method have allowed the optimisation of encapsulation processes to reach suitable encapsulation efficiencies. Nonetheless, the (double) emulsification step means that the encapsulation process occurs at random. The spontaneous emulsification (SE) solvent diffusion technique has been suggested as an improved method for controlled protein encapsulation. With SE, a more homogenous distribution of the encapsulated antigen is achieved, thus reducing the initial burst effect [139]. The SE method was validated by encapsulating bovine serum albumin (BSA): the in vitro release kinetics demonstrated a pulsatile-release of BSA and comparable antibody titres after a single-dose subcutaneous administration to BALB/c mice [144]. Technologies such as microfluidics and spray-drying have demonstrated the feasibility of more controlled encapsulation processes and may be critical in achieving the necessary encapsulation efficiencies for large scale vaccine manufacturing.

It has been demonstrated that the W/O/W method can be adapted to produce particles of desired size, however, it is quite difficult to achieve an extremely narrow size distribution (monodispersity). The size variation is largely due to the emulsification step, during which the particles have different droplet sizes, followed by variable precipitation rates during the solidification phase. The uniformity of particle size and distribution is also an important factor regarding the reproducibility and feasibility to scale up. Sieving the particles after emulsification can result in more uniform particle preparations. This is referred to as extrusion and achieved with Shirasu porous glass (SPG) beads [119,124,145]. Alternatively, encapsulation using novel microfluidic technologies may be a promising approach to achieve both adaptable particle sizes and a narrow size distribution. There remains a concern of translating laboratory-scale success to commercial-scale manufacture. Conventional emulsification methods may be inflated to a large scale of more than hundreds of litres per hour, however, there are numerous process parameters that need to be optimised individually [146]. This is an important aspect to take into consideration when validating novel antigen encapsulation technologies.

There is an increasing interest in developing core-shell particles with hydrophilic cores to maintain the microenvironment and therefore the conformation and biological integrity of the antigen [147]. They can be formulated using W/O/W method, microfluidics, or by StampEd assembly of polymer layers (SEAL), a recently reported microfabrication approach based on 3D printing [148]. Core:shell microparticles (25 µm) with an acceptable size distribution (span) of 1.4 were manufactured using the W/O/W method combined with ionotropic gelation to form a sodium alginate-based hydro-core (Figure 3a,b) (Lemoine et al., unpublished). Using fluorophore labelling and confocal imaging, the core:shell structure of these particles was demonstrated and a variable distribution of the alginate cores was observed. Comparatively, core:shell particles with different shell thicknesses were manufactured by a microfluidics method. Using two microfluidic flow-focusing designs consecutively, the core and shell size of the particles were modified independently to produce two different populations, namely “thin-” and “thick-” shell particles. Both populations were highly monodispersed, with the coefficient of variation less than 5% (Guyon et al., unpublished). Precise and identical size parameters across a population of particles could contribute to a sharper burst release of the payload, as particles are likely to exhibit the same behaviour if they are monodispersed. The identification of other significant factors affecting the release is also more tractable when the size of a particle batch is controlled.

## 7. Approaches to Encapsulation Using Microfluidics

Microfluidics involves the manipulation of fluid flows at a microscale, giving the fluids a laminar behaviour, and is used in specific applications such as droplet microfluidics. Thus, microfluidics can produce highly monodispersed droplets in a controlled and repeatable manner, a feature utilised in drug delivery, and in particular for vaccine encapsulation.

In microfluidics, droplets are generated by intersection designs, where two immiscible or partially miscible phases are put into contact, and subsequently produce droplets by the combined actions of shear stress, viscous forces, and interfacial tension. This process can be conducted in capillaries assembled coaxially in “co-flow” type intersections or in microfluidic chips, often made with polydimethylsiloxane (PDMS)—a silicon polymer poured onto a mould and hardened to form microfluidic chips—or with glass, where the “T-junction” or “flow-focusing” designs are embedded. Multiple emulsions are formed with the different fluids flowing through successive intersections, generating successive layers of the multiple emulsion (see Figure 4). These template multiple emulsions, either produced by capillary microfluidics [149,150], or chip microfluidics [151,152], are then converted into microcapsules.

### 7.1. Benefits of Using Microfluidics

As mentioned previously, the droplets generated by microfluidics exhibit an unmatched uniformity. The usual coefficient of variation (CV) of the droplet diameter is below 5% [153,154]. This property is essential for vaccine encapsulation as the uniform size not only ensures uniform antigen loading and dosing but also enables to assess the relationship between the particle size and the antigen release kinetics without the noise brought by polydispersity.

Microfluidics also demonstrate high encapsulation efficiencies (EE) as the emulsification process is not random but happens in a user-controlled way. Thus, Pessi et al. reported a 84% EE for BSA in polycaprolactone microcapsules produced with capillary microfluidic double emulsion [155]. Moreover, as the droplet generation occurs repeatably at a defined intersection, it minimizes the contact between the solvent phase and the antigen phase, while removing any agitation stress that takes place in classic batch emulsification methods, thus lowering the risks of loss of antigenicity.

Droplet microfluidic methods exhibit a user-friendly modularity, where the size and frequency generation can be optically monitored and changed by modifying the flow rates of the injected fluids in the microfluidic setup, as experimental law has been reported between the flow rates ratio and the size of the droplets produced [156]. This reduces the amount of post-production controls and potential waste by out-of-range production.

### 7.2. Towards Implementing Microfluidics for Vaccine Delivery Systems

Despite all the benefits of microfluidics outlined here, no direct use of this technology in the production of encapsulated vaccines for delayed delivery has been reported yet. Difficulties in setting up adequate microfluidics platform and processes could be the main reason. Indeed, microfluidics applications require both an accurate fluid flow controller, such as pressure pumps and connected reservoirs, or syringe pumps, and an imaging system, composed of a microscope, or a similar set of lenses and light, and a high-speed camera. These components are not frequently found in biology laboratories, and their setup and use also involves an engineering skill set.

Moreover, the microfluidic element itself necessitates manufacturing that has not been upscaled for mass production yet by the industry. Thus, capillary microfluidics systems are homemade, with many challenges to face in the precise assembly, while microfluidic chips are in most cases produced by soft lithography, involving clean room facilities and expertise.

The inability of microfluidics systems to achieve the yields of production needed by the drug manufacturing industry is often mentioned as one of the major drawbacks of microfluidics approaches. Indeed, although microfluidic droplet generation rate can reach several kilohertz, the total volume encapsulated is limited due to the inherent microscale aspect of the technology. However, parallelisation approaches, where multiple droplet microfluidic processes are performed simultaneously with the adequate designs, have been published and demonstrated more than acceptable throughputs [146,157,158]. Thus, Yadavali et al. reported a production rate of 277 g per hour of polycaprolactone microparticles using a silicon and glass parallelised microfluidic device [158].

Finally, the cost of vaccination must be kept in consideration, in particular when developing sophisticated technologies for vaccine formulation and delivery. The mobilisation of global funds for vaccine development, led by the WHO and dedicated organisations such as the Gavi Alliance, along with the scalability of methods and increasing the vaccine production capacity in the developing countries which significantly reduces the cost per dose [159], can help accelerate access to vaccines where they are most needed.

## 8. Conclusions

From the above considerations, it is evident that non-invasive vaccination systems are in development as well as are advanced delivery systems that may prove to be an essential part of promising multi-platform vaccine strategies. However, it has yet to be shown whether the presented technologies may provide reliable and cost-effective approaches to vaccination, yielding more efficient vaccines and improved patient compliance to immunisation programmes in both the developed and the developing countries.

## Figures and Tables

**Figure 1 vaccines-08-00304-f001:**
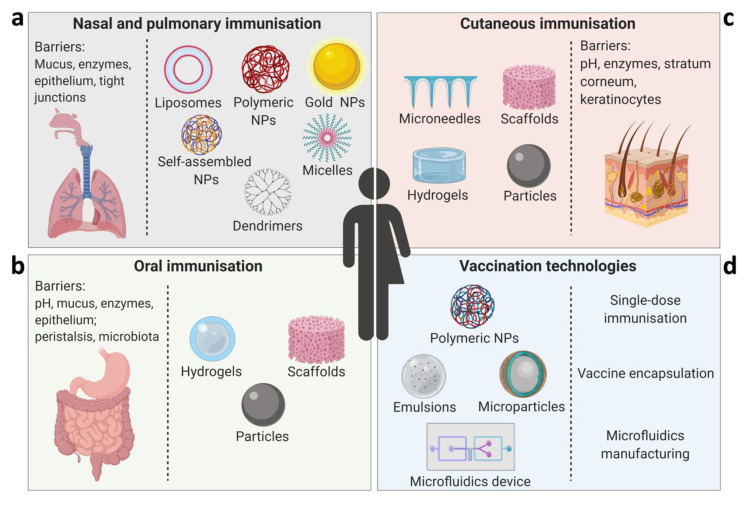
Advanced vaccination technologies and strategies for improving compliance and coverage. Key drug delivery systems to overcome barriers associated with each administration route are presented: (**a**) nasal and pulmonary immunisation using particulate delivery systems such as lipid-based systems (liposomes or nanocapsules), polymeric nanoparticles (NPs), gold NPs, self-assembled NPs (e.g., chitosan), dendrimers, and micelles; (**b**) oral immunisation using delivery systems such as hydrogels, scaffolds, and particles (nano- and microparticles); (**c**) cutaneous immunisation can be performed using microneedles, scaffolds, hydrogels, nano- and microparticles; (**d**) advanced technologies for improving vaccine manufacture and delivery, such as single-dose immunisation using polymeric NPs, and vaccine encapsulation using emulsions or microfluidics systems. Figure prepared using BioRender.

**Figure 2 vaccines-08-00304-f002:**
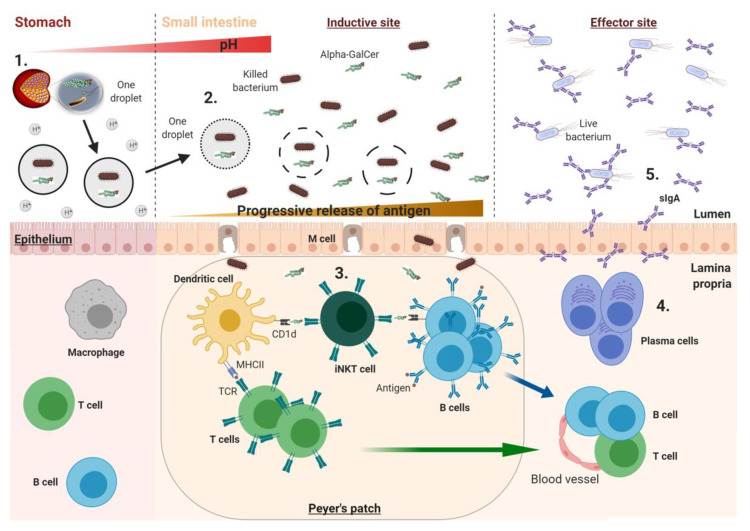
Proposed mechanism of the SmPill^®^ minispheres and the induction of intestinal vaccine-specific immune responses: 1. The enteric coating of SmPill^®^ minispheres remains intact in the acidic environment of the stomach protecting the payload; 2. On exiting the stomach and the passage into the increasing pH of the small intestine, the enteric coating begins to degrade, exposing the gelatine core and releasing the oil droplets containing the vaccine antigen (e.g., whole-cell killed bacteria) and the solubilised adjuvant (e.g., α-GalCer); 3. The payload is gradually released in the small intestine and the antigen/adjuvant can cross the intestinal epithelium (e.g., through M cells) where the presentation of processed whole-cell killed bacteria and α-GalCer by DCs to T cells and invariant natural killer T (iNKT) cells occurs, respectively. This leads to B cell activation; 4. B cells undergo affinity maturation, class switch recombination and differentiation into plasma cells, which enter into the circulation and home back to the lamina propria where antigen-specific IgA secretion occurs; 5. Upon infection with viable bacteria, sIgA transported into the intestinal lumen can neutralise the bacteria. Figure prepared using BioRender.

**Figure 3 vaccines-08-00304-f003:**
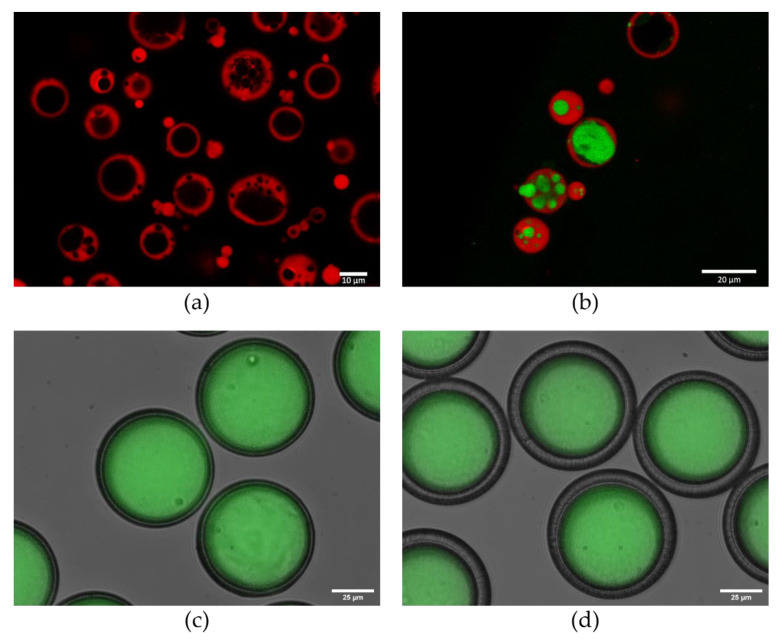
Core:shell microparticles prepared by two different manufacture methods. Confocal images of (**a**) the microparticles by the water-in-oil-in-water (W/O/W) method, core: sodium alginate 75–200 kDa (NovaMatrix^®^), shell: PLGA–rhodamine (50:50 30 kDa); (**b**) the microparticles by the W/O/W method, core: sodium alginate labelled with calcein, shell as in (**a**). Fluorescent microscopy images (40 × magnification) of (**c**) the thin shell microparticles by the microfluidics method, core: dextran-FITC 70 kDa, shell: PLGA resomer R502 (50:50 7–17 kDa); (**d**) thick shell microparticles by the microfluidics method (same parameters as in (**c**)).

**Figure 4 vaccines-08-00304-f004:**
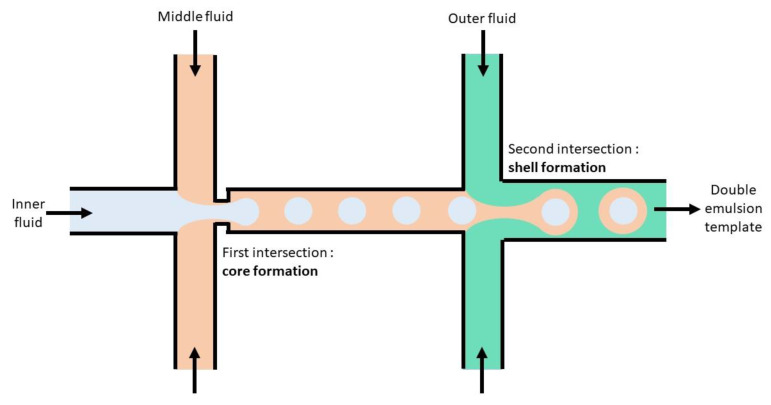
Double emulsion template formation by successive intersections in microfluidic chip.

**Table 1 vaccines-08-00304-t001:** Pulmonary and nasally delivered vaccines in clinical trials.

Study Title	Pathogen/ Condition	Vaccine	Administration Route	Status	Phase	Clinical Trials Gov Identifier
**Intrapulmonary**						
Phase 1 Clinical Trial of the Safety and Immunogenicity of an Adenovirus-Based TB Vaccine Administered by Aerosol	Tuberculosis	Ad5Ag85A	Pulmonary	Recruiting	I	NCT02337270
Investigating Immune Responses to Aerosol Bacillus Calmette–Guérin (BCG) Challenge in Healthy UK Adults	Tuberculosis	BCG Danish	Pulmonary	Recruiting	I	NCT03912207
ChAdOx1 85A Aerosol Versus Intramuscular Vaccination in Healthy Adults (TB039) (TB039)	Tuberculosis	ChadOx1 85A	Pulmonary	Recruiting	I	NCT04121494
**Nasal**						
Evaluating the Safety and Immune Response to a Single Dose of a respiratory syncytial virus (RSV) Vaccine in Infants and Children	RSV infection	RSV ΔNS2 Δ1313 I1314L	Nasal	Recruiting	I	NCT01893554
Safety and Immunogenicity of a RSV Vaccine in RSV-Seropositive Children and RSV-Seronegative Infants and Children	RSV infection	D46cpΔM2-2 vaccine	Nasal	Active, not recruiting	I	NCT02601612
Evaluating the Infectivity, Safety, and Immunogenicity of the Recombinant Live-Attenuated RSV Vaccines in RSV-Seronegative Infants 6 to 24 Months of Age	RSV infection	RSV ΔNS2/Δ1313/I1314L and RSV 276	Nasal	Active, not recruiting	I	NCT03227029
Evaluating the Infectivity, Safety, and Immunogenicity of a RSV Vaccine in RSV-Seropositive Children and RSV-Seronegative Infants and Children	RSV infection	RSV 6120/∆NS2/1030s	Nasal	Recruiting	I	NCT03387137
Evaluating the Infectivity, Safety, and Immunogenicity of the Recombinant Live-Attenuated RSV Vaccines in RSV-Seronegative Infants and Children 6 to 24 Months of Age	RSV infection	RSV ΔNS2/Δ1313/I1314L and RSV 276	Nasal	Active, not recruiting	I	NCT03422237
A Study Assessing Colonisation and Immunogenicity after Nasal Inoculation with *N. lactamica* and Eradication on Day 4 or 14	Meningitis	*Neisseria lactamica*	Nasal	Recruiting	Not applicable	NCT03549325
Evaluating the Infectivity, Safety and Immunogenicity of RSV Vaccines in RSV-Seropositive Children and RSV-Seronegative Infants and Children	RSV infection	RSV 6120/∆NS1 and RSV 6120/F1/G2/∆NS1	Nasal	Recruiting	I	NCT03596801
Mucosal and Systemic Immunity after Viral Challenge of Healthy Volunteers Vaccinated with Inactivated Influenza Vaccine via the Intranasal Versus Intramuscular Route	Influenza	Flucelvax: Inactivated influenza vaccine	Nasal	Recruiting	II	NCT03845231
Safety and Immunogenicity of a Single Dose of the Recombinant Live-Attenuated RSV Vaccines or placebo, delivered as nose drops to RSV-Seronegative Children 6 to 24 Months of Age	RSV infection	RSV ΔNS2/Δ1313/I1314L, RSV 6120/∆NS2/1030s, and RSV 276	Nasal	Recruiting	I/II	NCT03916185
Nasal and Systemic Immune Responses to Nasal Influenza Vaccine (Flu-M3)	Influenza	Live attenuated influenza vaccine	Nasal	Active, not recruiting	Not applicable	NCT04110366
A Controlled Study to Assess Safety, Colonisation and Immunogenicity of Reconstituted Lyophilised Neisseria lactamica (Lac5-Nasal)	Meningitis	Lyophilised Neisseria lactamica	Nasal	Recruiting	Not applicable	NCT04135053
A Safety and Immunogenicity of Intranasal Nanoemulsion Adjuvanted Recombinant Anthrax Vaccine in Healthy Adults (IN NE-rPA)	Anthrax	BW-1010: a nanoemulsion adjuvanted recombinant protein	Nasal	Recruiting	I	NCT04148118
Live-Attenuated Influenza Vaccine as a Nasal Model for Influenza Infection	Influenza	Flumist quadrivalent nasal vaccine	Nasal	Not yet recruiting	IV	NCT04164212

HSP, heat shock protein.

**Table 2 vaccines-08-00304-t002:** Delivery systems for pulmonary and nasally delivered vaccines.

Delivery System	Pathogen/Antigen	Administration	Animal Model	Immunity Type Generated	Reference
Liposomes	*Mycobacterium tuberculosis* (*Mtb*) H56 antigen	Pulmonary	Mice	Th1; Th17; IgA; IgG	[29]
ISCOMs	Human T cell lymphotropic virus type 1	Nasal	Mice	Th1; IgA; IgG	[30]
Chitosan	Dengue virus	Nasal	Mice	CD8^+^ T cells; IgA; IgG	[31]
γ-polyglutamic acid	Group A *Streptococcus*	Nasal	Mice	IgA; IgG	[32]
Hyaluronic acid	Influenza hemagglutinin	Nasal	Mice, Rabbits, Micro-pigs	IgA; IgG	[33]
Pullulan	*Streptococcus pneumoniae*	Nasal	Macaques	Th2; Th17; IgA; IgG	[34]
**Synthetic polymer-based particles**					
PLGA	*Chlamydia trachomatis*	Nasal	Mice	Th1; IgA; IgG	[35]
PEI	H9N2 Influenza	Nasal	Mice	Th1; CD8^+^ T cells; IgA; IgG	[36]
PCL	Hepatitis B	Nasal	Mice	IgA; IgG	[37]
PPS	*Mtb*	Nasal	Mice	Th1; Th17	[38]
**Inorganic particles**					
Gold particles	H3N2 hemagglutinin	Nasal	Mice	Th1; CD8^+^ T cells; IgA; IgG	[39]
Aluminium particles	Ovalbumin	Nasal	Rats	IgA; IgG	[40]
Calcium phosphate particles	Chimeric dengue virus serotype 2	Nasal	Mice	IgA	[41]
Silica-based particles	Foot and mouth disease virus	Nasal	Guinea pigs	IgA; IgG	[42]
Carbon nanoparticles	Ovalbumin	Nasal	Mice	Th1; CD8^+^ T cells	[43]
**Infectious materials**					
Recombinant bacteria	Lactobacillus plantarum vector for Mtb	Nasal	Mice	Th1; IgA	[44]
Recombinant virus	Influenza virus vector for respiratory syncytial virus	Pulmonary; Nasal	Mice	CD8^+^ T cells	[45]
Outer membrane vesicles (OMV)	Bacteroides thetaiotaomicron OMV for Yersinia pestis V and F antigen	Nasal	Mice	IgA; IgG	[46]
Emulsions	Helicobacter pylori	Nasal	Mice	Th1; IgA; IgG	[47]
VLPs	Influenza VLPs	Nasal	Mice	Th1; IgA; IgG	[48]

HSP, heat shock protein; Ig, immunoglobulin; ISCOMs, immunostimulatory complexes; PCL, poly ε-caprolactone; PEI, polyethyleneimine; PLGA, poly (lactic-co-glycolic acid); PPS, polyphenylene sulphide; Th, helper T cells; VLPs, virus-like particles.

**Table 3 vaccines-08-00304-t003:** Immunopotentiators for pulmonary and nasally delivered vaccines.

Immunopotentiator	Pathogen/Antigen	Administration	Animal Model	Immunity Type Generated	Reference
**Bacterial TLR agonists**					
Lipopeptides: TLR-1/2 agonists	*Mycobacterium tuberculosis*	Nasal	Mice	Th1; Th17	[49]
Lipopolysaccharide: TLR-4 agonist	Human T cell lymphotropic virus type 1	Nasal	Mice	Th1; IgA; IgG	[30]
Peptidoglycan: TLR-2/4 agonists	Respiratory syncytial virus	Nasal	Mice	Th1; Th2	[50]
Flagellin: TLR-5 agonist	Influenza A virus	Nasal	Mice	Th1; CD8^+^ T cells; IgA; IgG	[51]
CpG DNA: TLR-9 agonist	Foot and mouth disease virus	Nasal	Guinea pig	IgA; IgG	[42]
**Viral TLR agonists**					
Double stranded RNA: TLR 3 agonist	Human parainfluenza virus type 3 virus	Nasal	Mice; Cotton rats; Pigs	Th1; IgA	[52]
Guanosine analogues: TLR-7/8 agonists	*Entamoeba histolytica*	Nasal	Mice	Th1; Th17; IgA; IgG	[53]
STING agonist: Cyclic dinucleotide GMP–AMP	H1N1, H3N2, H5N1, H7N9 Influenza	Nasal	Mice; Ferrets	Th1; CD8^+^ T cells; IgA; IgG	[54]
**Cytokines**					
Type I Interferons (IFN)	Influenza	Nasal	Mice	IgA; IgG	[55]
IFN-γ	*Yersinia pestis*	Nasal	Mice	IgA; IgG	[56]
GM-CSF	HIV-1	Nasal	Mice	IgA; IgG	[57]
IL-12	HIV	Nasal	Mice	Th1; CD8^+^ T cells; IgA; IgG	[58]
IL-15	Simian immunodeficiency virus	Pulmonary	Mice	Th1; CD8^+^ T cells; ADCC	[59]
IL-18	HIV	Nasal	Mice	Th1; CD8^+^ T cells	[60]
FLT-3 ligand	*Chlamydia abortus*	Nasal	Mice	Th1; IgA; IgG	[61]

ADCC, antibody-dependent cell-mediated cytotoxicity; CpG, cytidine-phosphateguanosine; FLT-3, Fms-Like tyrosine kinase 3; GM-CSF, granulocyte-macrophage colony-stimulating factor; GMP-AMP, guanosine monophosphate–adenosine monophosphate; HIV, human immunodeficiency virus; HN, hemagglutinin and neuraminidase; IL, interleukin; STING, stimulator of interferon genes; TLR, toll-like receptor.

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
