# Peer review of "Technological Approaches for Improving Vaccination Compliance and Coverage"

_vaccines, 2020, doi:10.3390/vaccines8020304_

Round 1

Reviewer 1 Report

This review article is very well written and describe the advances in formulation and delivery of vaccines for improving compliance and coverage. It is a very important subject in the vaccine research on increasing  compliance to get wide scale immunity in the population. Authors have covered almost every topic and discuss the new technologies in vaccine formulation. I have following suggestions:

Suggestion:

  1. Could authors also add more information regarding use of antibiotics in vaccine formulation.
  2. Authors can also discuss about any advances in chemical stabilizer used in vaccine formulation.  

Comment:

Cost of vaccination is an important issue and it affects the compliance particularly in developing countries. There need to be some discussion on what public/private partnership can do to maintain the cost of new technologies based vaccine low for improving compliance and coverage.  

Author Response

Reviewer 1

This review article is very well written and describe the advances in formulation and delivery of vaccines for improving compliance and coverage. It is a very important subject in the vaccine research on increasing  compliance to get wide scale immunity in the population. Authors have covered almost every topic and discuss the new technologies in vaccine formulation. I have following suggestions:

Suggestion:

  1. Could authors also add more information regarding use of antibiotics in vaccine formulation.

We thank the reviewer on the positive assessment of the manuscript.

This suggestion relates to an important aspect of vaccine uptake from the point of view of vaccine hesitancy through concerns over the use (and any residual amounts) of antibiotics in vaccine manufacture. However, vaccine formulation with respect to vaccine active ingredients and excipients is beyond the scope of this review, as is vaccine hesitancy as a social phenomenon, and we feel that addressing them would detract from the technological theme of the manuscript. We have removed the term ‘formulation’ from the list of keywords to avoid inaccuracy.

  1. Authors can also discuss about any advances in chemical stabilizer used in vaccine formulation.  

We thank the reviewer for highlighting this issue. Advances in chemical stabilisation of vaccines are now included in the context of developing thermostable formulations (lines 169-183) and oral vaccines (lines 225-229 and 292-300).

Comment:

Cost of vaccination is an important issue and it affects the compliance particularly in developing countries. There need to be some discussion on what public/private partnership can do to maintain the cost of new technologies based vaccine low for improving compliance and coverage.  

This is another important point which we have addressed in the revised document (lines 628-633).

Reviewer 2 Report

The review article by Lemoine et al presents a comprehensive overview of the various vaccination techniques and strategies that are either in use or under development. The review overall is very well written and is well organized. There are few specific suggestions that will improve this article.

The authors have included examples and outcome in the PLGA polymer vaccine delivery systems that helps the reader understand the point clearly. The authors should include similar pattern for other strategies that are mentioned throughout the review article. For eg., line 96-102: nasal and pulmonary vaccination techniques are presented well. However, including some examples of the vaccines and their outcome will allow the reader to grasp the significance without searching through other articles.

The review mentions the interesting role of the M cells in transporting antigens across the epithelium. Are there any targeted strategies that can be used to better engage these cells?

Table 2 footnotes include “HIV, human immunodeficiency virus; HN, hemagglutinin and neuraminidase” but are not mentioned in the table.

Line 96: “Intranasal administration of a vaccine allows the induction a strong systemic and local immune”: should be “Intranasal administration of a vaccine allows the induction of a strong systemic and local immune”

Line 97: “nasal cavity is relatively weak, comparing to the oral route” should be “nasal cavity is relatively weak, compared to the oral route”. Thorough proof-reading is required.

Author Response

Reviewer 2

The review article by Lemoine et al presents a comprehensive overview of the various vaccination techniques and strategies that are either in use or under development. The review overall is very well written and is well organized. There are few specific suggestions that will improve this article.

The authors have included examples and outcome in the PLGA polymer vaccine delivery systems that helps the reader understand the point clearly. The authors should include similar pattern for other strategies that are mentioned throughout the review article. For eg., line 96-102: nasal and pulmonary vaccination techniques are presented well. However, including some examples of the vaccines and their outcome will allow the reader to grasp the significance without searching through other articles.

We thank the reviewer for this suggestion. We have included some examples of vaccines and their outcome in the text in several places (lines 99-105, 180-185).

The review mentions the interesting role of the M cells in transporting antigens across the epithelium. Are there any targeted strategies that can be used to better engage these cells?

We have now included a section on M cell targeting (lines 234-256).

Table 2 footnotes include “HIV, human immunodeficiency virus; HN, hemagglutinin and neuraminidase” but are not mentioned in the table.

We thank the reviewer for noticing this mistake, which has been now corrected (Table 2 footnotes).

Line 96: “Intranasal administration of a vaccine allows the induction a strong systemic and local immune”: should be “Intranasal administration of a vaccine allows the induction of a strong systemic and local immune”

We thank the reviewer for noticing this mistake which has been now corrected (line 98).

Line 97: “nasal cavity is relatively weak, comparing to the oral route” should be “nasal cavity is relatively weak, compared to the oral route”. Thorough proof-reading is required.

This has now been corrected and we have proof-read the manuscript, correcting any additional typographical errors.